# Isolation of Aroma-Producing *Wickerhamomyces anomalus* Yeast and Analysis of Its Typical Flavoring Metabolites

**DOI:** 10.3390/foods12152934

**Published:** 2023-08-02

**Authors:** Jing Zhang, Yiguo He, Liguo Yin, Rong Hu, Jiao Yang, Jing Zhou, Tao Cheng, Hongyu Liu, Xingxiu Zhao

**Affiliations:** 1College of Bioengineering, Sichuan University of Science and Engineering, Yibin 644000, China; 2Solidstate Fermentation Resource Utilization Key Laboratory of Sichuan Province, Yibin University, Yibin 644000, China

**Keywords:** Sichuan paocai, mixed fermentation, SPME-GC–MS analysis, volatile flavor components

## Abstract

In this study, 21 strains of aroma-producing yeast were isolated from Sichuan paocai juice of farmers in western, eastern and southern Sichuan. One strain, Y3, with the best aroma-producing characteristics, was screened using an olfactory method and a total ester titration method, and was identified as *Wickerhamomyces anomalus*. The total ester content of Y3 fermentation broth was as high as 1.22 g/L, and there was no white colonies or film on the surface. Meanwhile, the Y3 strain could tolerate 14% salt concentration conditions and grow well in a pH range of 3–4. Through sensory analysis, the fermented mustard with a ratio of *Lactiplantibacillus plantarum* to Y3 of 1:1 showed the highest overall acceptability. Ethyl acetate with its fruit and wine flavor was also detected in the fermented Sichuan paocai juice with a mixed bacteria ratio of 1:1, analyzed with SPME-GC–MS technology, as well as phenylethyl alcohol, isobutyl alcohol, isothiocyanate eaters, myrcene and dimethyl disulfide. These contributed greatly to the unique flavor of Sichuan paocai. In general, *Wickerhamomyces anomalus* Y3 enhanced the aroma of the fermented Sichuan paocai.

## 1. Introduction

Traditional Sichuan paocai (SCP) is used to soak fresh vegetables, such as cabbage, garlic and ginger in 5–10% salt solution, and ferments by naturally adherent microorganisms on the vegetables, mainly including lactic acid bacteria, acetic acid bacteria and yeast [1]. SCP has a crisp and tender fragrance; it refreshes the appetite, and promotes digestion, has antibacterial characteristics and prevents hypertension [2,3]. During the fermentation process, SCP produces organic acids, ketones, aldehydes, esters and other flavor substances, which give the SCP a unique flavor and mouthfeel. The aromatic yeast plays an extremely important role in the formation of these flavors.

Aroma-producing yeast refers to yeasts that can produce aromatic substances during their growth and reproduction. These yeasts mainly produce ethyl acetate and other ester compounds, including *Hansenula* [4], *Candida*, *Pichia anomalous* [5], and so on. Currently, aroma-producing yeast is not only used in liquor brewing, but is also widely used in the brewing research of soy sauce [6,7], vinegar [8], rice wine [9], etc.

Currently, the research on SCP fermentation is mostly focused on lactic acid bacteria [10,11], and there have been a few studies on the isolation and screening of yeast strains with high aroma-producing performance and their application in SCP production.

This study aimed to isolate and identify aroma-producing yeast from SCP juice, and to study the related properties of the strains. We studied the sensory properties and aroma compounds of SCP fermented by *Lactiplantibacillus plantarum* (preserved by our laboratory) and aroma-producing yeast in different proportions. The aim is to improve the quality of SCP and broaden the application range of aroma-producing yeast in its production.

## 2. Materials and Methods

### 2.1. Materials

The SCP liquid was provided by farmers in western Sichuan (Qionglai), eastern Sichuan (Nanchong) and southern Sichuan (Yibin). The peptone and yeast extract powder were purchased from Beijing Aobo Biotechnology Co., Ltd. (Beijing, China) Concentrated sulfuric acid, glucose, maltose, sodium hydroxide, and absolute ethanol were purchased from Sinopharm Chemical Reagent Co., Ltd. (Shanghai, China) Sodium carboxymethyl cellulose was purchased from Chengdu Kelong Chemical Co., Ltd. (Chengdu, China).

### 2.2. Isolation and Screening of Aromatic Yeast

Three portions of SCP stock were incubated in enrichment medium (peptone 6 g, glucose 6 g, yeast paste 3 g, distilled water 300 mL, natural pH, chloramphenicol 0.1 g) at 28 °C and 120 r/min for 24 h, and then a 10-fold dilution gradient method was used. The strains were cultured in red Bengal solid medium (peptone 2.5 g, glucose 5 g, potassium dihydrogen phosphate 0.5 g, magnesium sulfate 0.25 g, agar 10 g, Bengal red solution 0.033 g, chloramphenicol 0.01 g) [12] at 28 °C for 48 h. Colonies of different morphologies were selected from the medium and cultured in YPD medium (peptone 5 g, glucose 5 g, yeast paste 2.5 g, agar 5 g, distilled water 250 mL) for 2 to 3 times in line, in order to improve the purity of the strain. The purified strain was refrigerated at 4 °C.

Yeast strains with obvious flavor were screened using an olfactory method [9]. After the purified yeast was activated in YPD liquid medium, it was diluted with sterile water. The concentration of yeast cell suspension was adjusted using ultraviolet spectrophotometry and a blood plate counting method. The optical density OD_560_ nm value was adjusted to about 0.5, and the yeast count reached 10^6^ CFU/mL. A volume of 0.2 mL was transferred and coated on PDA medium (100 g of potato and 500 mL of distilled water boiled for 30 min, filtrate, glucose 10 g, agar 10 g), then cultured at 28 °C for 3 days, smelling the odor produced in the PDA medium. The yeast strains with obvious aroma were screened out.

The yeast suspension obtained via the olfactory method was prepared into a yeast suspension to adjust the optical density OD_560_ nm value to 0.5, and inoculated in 100 mL of YPD liquid medium to 3%. The total ester content of the screened strains was determined after 4 days of shaking culture at 28 °C and 120 r/min [6]. The 50.00 mL of fermentation broth was accurately drawn into a 250 mL conical flask, and 2 drops of phenolphthalein indicator were added. The fermentation broth was titrated to a slightly red color with 0.1 mol/L NaOH standard solution. The titrated fermentation broth was transferred to a 250 mL round-bottom flask, and 25.00 mL of 0.1 mol/L NaOH standard solution was added. The condensation tube was connected, saponified in a boiling water bath for 0.5 h, cooled down to room temperature, and completely transferred to a 250 mL iodometric flask. The fermentation broth was immediately titrated with a 0.1 mol/L H_2_SO_4_ standard solution until the slightly red color disappeared. The amount of H_2_SO_4_ standard solution was recorded, and the total ester content in the fermentation broth was calculated as follows:$$\mathrm{Total}\;\mathrm{ester}\;(g/L)=\frac{c\left(\mathrm{NaOH}\right)\times 25.00-c\left(H_{2}{\mathrm{SO}}_{4}\right)\times V}{50.0}\times 0.08812\times 1000$$
where V is the volume of H_2_SO_4_ standard titration solution consumed in the second titration; 0.08812 is the mass of ethyl acetate equivalent to 1.00 mL NaOH standard titration solution.

### 2.3. Performance Determination of Aroma-Producing Yeast Strains

The yeast strains were inoculated in liquid YPD medium (10^6^ CFU/mL) and added with 4%, 6%, 8%, 12%, 14% and 16% NaCl, respectively, to detect and analyze the growth conditions under different NaCl concentrations. The effects of different pH values on yeast growth were observed in YPD at pH 2, 2.5, 3.0, 3.5, 4.0, 4.5, 5.0, 5.5, 6.0 and 6.5. The optical density (OD) of each yeast suspension was measured at 560 nm after 48 h of incubation at 28 °C. The white film on the surface was observed in the YPD medium after 48 h of culture.

The DNS (3,5-dinitrosalicylic acid) method was used to determine the cellulose and pectinase produced by the aroma yeast [13,14]. The yeast fermentation broth was centrifuged at 4000 r/min for 10 min, and the supernatant was taken as the crude enzyme solution. The standard curve was plotted with different dilutions of 1 mg/mL glucose solution and galacturonic acid solution, and the OD value was measured at 540 nm. For the analysis of enzyme activity, 1% CMC-Na (sodium carboxymethyl cellulose) solution and a 1% pectin solution prepared with disodium hydrogen phosphate-citric acid at pH 4.8 were used as substrates, and 0.5 mL of crude enzyme solution was added to each of 4 tubes (1 blank tube and 3 sample tubes); the blank tubes were inactivated in a boiling water bath for 10 min, and 1.5 mL of substrate solution was added to each one and reacted in a water bath at 45 °C for 45 min. Each tube received 3 mL of DNS reagent, then cooled to room temperature after 10 min in a boiling water bath, and fixed to 15 mL with distilled water. The OD value was measured at 540 nm using a blank tube as the control. Then, the enzyme activity was calculated. The international definition of enzyme activity units is as follows: an enzyme activity unit is defined as the amount of 1 mL of enzyme solution that decomposes the substrate to produce 1 μg of glucose or 1 μg of galacturonic acid in 1 min.

### 2.4. Identification of Aroma-Producing Yeast

#### 2.4.1. Morphological Observation

Each yeast strain was inoculated on a YPD plate and cultured at 28 °C for 2 days to observe the colony characteristics. Then, the cell morphology was examined using a microscope.

#### 2.4.2. Molecular Biological Identification

The yeast strains were sent to Shanghai Jie Li Biotechnology Co., Ltd. (Shanghai, China) for sequencing. The sequencing results were submitted to the GenBank database of the National Center for Biotechnology Information, and a basic local alignment search tool (BLAST) comparison search was performed. The 26S rDNA gene sequence of the model strain with the highest homology was selected for multiple sequence comparison using Clustalx1.83 software, and the phylogenetic tree was constructed using the neighbor joining (NJ) method in Mega7.0 software [15].

### 2.5. Fermentation of SCP

Mustard is bitter and spicy, and is often used in the processing and production of SCP. Fresh mustard was selected as SCP raw material. The damaged parts were washed and removed, and the rest was cut into small pieces (4–6 cm), put into an SCP fermentation tank; spices were added, including garlic, ginger, pepper and Chinese prickly ash, etc. Then, the same amount of cold boiled water (salt content 6%) was added. The *Lactiplantibacillus plantarum* and Y3 yeast preserved in our laboratory were added to fresh mustard at ratios of 1:1, 2:1 and 3:1 (inoculation amount of 1%) for mixed fermentation, natural fermentation, and mustard without strains was used as a control. The jars were sealed with water and fermented at room temperature for 7 days. The improvement effect of aroma-producing yeast on the quality and flavor of SCP was evaluated using sensory evaluation and SPME-GC–MS analysis. 

### 2.6. Liquid Aroma Production Experiment of Aroma-Producing Yeast

The aroma components in the SCP juice were extracted via headspace solid-phase microextraction (SPME) and analyzed with gas chromatography–mass spectrometry (GC–MS) [9].

A 5 mL sample with 5 μL of 4-methyl-2-pentanol methanol solution (0.4 μg/mL, internal standard compound) and 1 g of NaCl were added into a 15 mL headspace vial, and the samples were equilibrated at 50 °C for 10 min. The pre-aged extraction head was inserted into the extraction vial for equilibration for 30 min and adsorption for 10 min. The sample was desorbed at 230 °C for 3 min for gas phase detection. The GC–MS setup procedure used was the following: the temperature was increased by 40 °C and kept for 2 min. The temperature was raised 5 °C/min to 180 °C, and then raised to 230 °C at 25 °C/min for 5.5 min. The inlet temperature was 230 °C, the detector temperature was 230 °C and the carrier gas was high-purity helium. The electron collision energy was 70 eV, the resolution was 1000, and the ion source was set to 250 °C. Electron impact (EI) mass spectra were recorded in the 50–550 amu range.

### 2.7. Sensory Evaluation

The aroma, color, taste, texture and overall acceptability of SCP were evaluated by 10 trained evaluators (5 males and 5 females, aged 22–45 years) from Sichuan University of Science & Engineering. The samples were presented to the evaluators in a random order. The sensory scoring used was the following: unsatisfactory (1 point), general (2 points), medium (3 points), good (4 points), excellent (5 points) [16].

### 2.8. Data Analysis

Origin 2021b software (OriginLab Co., Northampton, MA, USA) was used for drawing processing. The significance of the data was determined using one-way analysis of variance (ANOVA), followed by Duncan’s multiple range test, via SPSS 26.0 (SPSS Inc., Chicago, IL, USA), with a 95% confidence level, meaning that differences were considered to be statistically significant when *p* < 0.05.

## 3. Results and Discussion

### 3.1. Screening of Aroma-Producing Yeast

Twenty-one strains of yeast were isolated and purified from the pickle juices of western, eastern and southern Sichuan, and five yeast strains with strong aroma were initially screened using the olfactory method, which were numbered Y3, Y4, Y5, Y6 and Y7 in sequence. It can be seen from Table 1 that the aroma characteristics of the five strains were roughly the same, but there was a difference in the intensity of aroma production. The Y3 and Y5 strains produced a relatively strong ester aroma and wine aroma on the solid medium, indicating that the aroma production performance was better. The Y4, Y6 and Y7 solid medium produced ester and a wine aroma that was lighter, indicating that the aroma production performance was poor.

The total ester determination results of the five strains are summarized in Figure 1. The total ester contents of the five strains were all above 1 g/L, and the total contents in Y3 and Y5 were 1.22 g/L and 1.2 g/L, respectively. The different types and activities of enzymes in different yeasts resulted in different esterification abilities of the five strains under the same fermentation conditions [17]. Combined with the results of preliminary screening and rescreening, the Y3 and Y5 strains were selected as the flavor-producing strains for further study.

### 3.2. Determination of Salt and Acid Resistance

We investigated the salt and acid resistance of the five yeast strains by simulating the salt concentration and pH value in the SCP fermentation environment. During the fermentation process of SCP, yeast is often subject to various environmental stresses, including salt, acid, etc. [18]; especially high concentrations of salt solution on the cell membrane of yeast result in functional protein dehydration, loss of original utility, and changes in cell membrane permeability, thereby inhibiting the growth of yeast cells; higher salt concentrations may even kill yeast cells. Meanwhile, high concentrations of salt can also cause the formation of reactive oxygen species (ROS) in cells, thus affecting changes in intracellular redox homeostasis; maintaining redox homeostasis is considered to be one of the important factors for cell salt tolerance [19].

The salt tolerance of yeast was determined by its growth under different NaCl concentrations. It can be seen from Figure 2a that with increasing NaCl concentrations, the growth of the yeast gradually decreased. Among them, the Y4, Y5 and Y7 strains stopped growing after the NaCl concentration was 8%, the Y6 strain tended to zero after the NaCl concentration was 12%, and the Y3 strain tended to zero after the NaCl concentration was 14%. It shows that with increasing salt concentrations, the tolerance of yeast can be reduced; SCP is a food with a high salt content, so the normal growth of aroma-producing yeast strains under high salt content is the basic prerequisite for its application in high-salt SCP fermentation.

In addition, with the extension of fermentation time, lactic acid fermentation with *lactic acid bacteria* as the main dominant bacteria will produce large amounts of lactic acid, which will reduce the pH of the fermentation system to 3.2~3.6 [20]. The poor tolerance of yeast to acidity may lead to slow fermentation speeds in the early stages of fermentation, and even early termination of fermentation in severe cases. According to the pH range of SCP, the growth of yeast under different pH conditions was determined. It can be seen from Figure 2b that the growth of the Y6 and Y7 strains showed a downward trend after pH 3.0; the growth of Y3 strains showed an upward trend within pH 3~4; and the growth of the Y4 and Y5 strains showed an upward trend within pH 3~4, but the growth was lower than that of the Y3 strains.

### 3.3. White Colony or Buffy Coat Formation Assay

According to the research [21], in the process of SCP fermentation, the reproduction of some yeasts or harmful microorganisms will affect the quality of SCP, which is manifested in the production of odor and the formation of white colonies or films on the surface of the product. From Table 2, it can be seen that among the five yeast strains with better aroma-producing performance screened from pickle juice, the Y6 strain produced white film on the surface after 1 week of static culture, and the formation of white colonies on the surface of the SCP may be due to fermentation temperature or resistance to acidic substances [21], indicating that the Y6 strain was not suitable for SCP fermentation.

### 3.4. Determination of Cellulose and Pectinase

The fresh vegetables used as the raw materials of SCP are rich in cellulose and pectin, which make the SCP taste crisp and tender. Crispness is an important indicator of the quality of paocai fruit and vegetable matter. However, some yeasts and harmful microorganisms can not only form a white film on the surface of SCP, but also soften the product, which seriously affects its edibility and economic value. Therefore, strains that produce no or less cellulase and pectinase in the SCP fermentation system should be selected as much as possible. It can be seen from Table 3 that the five yeast strains we screened did not produce cellulase and pectinase, which indicated that the five yeast strains could be artificially inoculated into SCP to improve its brittleness.

### 3.5. Y3 Yeast Identification Results

#### 3.5.1. Morphological Characteristics of the Strain

Five strains of yeast with strong aroma were screened from the farm SCP juice using a smelling method. After a series of experiments, the Y3 yeast stood out because of its excellent ability to produce total esters, good salt and acid tolerance, and no production of white film, cellulose and pectinase. The strains that were not suitable for SCP fermentation were eliminated. Therefore, the Y3 yeast was selected for molecular identification and subsequent fermentation flavor analysis.

As can be seen in Figure 3a, strain Y3 had large and thick colonies on YPD medium, with a bulging center and relatively neat edges, and a thicker, opaque, creamy white appearance with a pleasant wine flavor. As can be seen from the light microscope images in Figure 3b, the yeast cells were ovoid and round, with multilateral outgrowth.

#### 3.5.2. Molecular Biological Identification of Y3 Strain

The 26S rDNA D1/D2 region sequence of yeast strain Y3 was queried against the NCBI database. A phylogenetic tree for related strains was constructed according to strain similarities (Figure 4). Strain Y3, *W. anomalus* MK267769.1 and GQ280811.1, clustered into a branch with the closest genetic relationship; the homology was 100%, which identified strain Y3 as *W. anomalus*. Currently, abnormal *W. anomalus* yeast is one of the most commonly isolated flavor-producing yeast. *W. anomalus* and *Pichia anomala* (synonym for abnormal *W. anomalus* yeast) [22] are important yeast strains for Baijiu brewing and traditional fermented food [23].

Currently, abnormal *W. anomalus* yeast is one of the most commonly isolated flavor-producing yeast. *W. anomalus* and *Pichia anomala* (synonym for abnormal *W. anomalus* yeast) [22] are important yeast strains for Baijiu brewing and traditional fermented food [23]. The study of *W. anomalus* yeast in SCP has not been extensively reported, particularly regarding the volatile aroma of specific strains and the sensory quality of SCP. Therefore, we chose to conduct mixed fermentation of *W. anomalus* Y3 and Lactiplantibacillus plantarum to investigate how the use of *W. anomalus* yeast affects the flavor and sensory quality of SCP.

### 3.6. Liquid Aroma Production Experiment of Aroma-Producing Yeast

The basic major volatile components were roughly the same in the naturally fermented and mixed fermented SCP juice (Table 4), with a total of 62 volatile compositions, including 10 esters, 7 aldehydes, 20 alcohols, 4 alkenes, 5 alkanes, 1 disulfide compound, 2 azole compounds, 2 amide compounds, 2 amino compounds and 6 others. The aroma components of natural fermentation included 41.17% esters, 2.86% aldehydes, 4.8% alcohols, 1.1% olefins, 2.03% amino compounds and 17.58% others. However, the contents of total esters and total alcohols in SCP juice fermented with yeast and *Lactiplantibacillus plantarum* increased significantly to 41.927%, 76.639%, 72.21%, 26.453%, 7.793% and 5.43%, respectively. Esters and alcohols are the main fermentation products of SCP, and play an important role in its flavor, depending on the type of compound and its concentration.

Although the total ester content in the fermentation group with the mixed bacteria ratio of 1:1:1 was much lower than that in the fermentation groups with the mixed bacteria ratios of 2:1and 3:1, ethyl acetate was detected in the fermentation group with the mixed bacteria ratio of 1:1. The formation of ethyl acetate was mainly due to the role played by yeasts, which, at low levels of *Lactobacillus*, produced lactic acid (homolactic fermentation) at levels that were comparable to the ethanol produced by the yeasts, which combined to produce ethyl acetate. Ethyl acetate can produce a strong ether flavor, a clear and slightly fruity flavor and liquor flavor, which has an important contribution to the flavor of SCP [24].

In addition, in the mixed fermentation group of 1:1, the content of phenylethyl alcohol significantly increased, which could endow the fermented samples with a honey and rose aroma. A high concentration of phenylethyl alcohol is usually detected in Chinese rice wine [25]. At the same time, in the 1:1 fermentation group was also detected the weak bitter allyl alcohol smell of isobutyl alcohol.

Hierarchical clustering of different fermented juices was performed, and the results are displayed in the heatmap in Figure 5. All of the samples were divided into four categories based on the vertical direction of the heatmap. As can be seen from the graph, the aroma composition varies considerably between the different fermented SCP juices, but one of their main products was esters with floral and fruity aromas [26]. Meanwhile, allyl isothiocyanate and phenylethyl isothiocyanate were detected in all four groups of the fermented SCP juices. These SCP juices are produced by vegetables (such as ginger, pepper, garlic, etc.) containing a large amount of volatile flavor substances. They have a green aroma and a strong irritating odor, which constitutes the unique flavor of SCP [27]. In addition, isobutyl isothiocyanate and 3-butene isothiocyanate were not detected in the naturally fermented vegetable juice, indicating that mixed fermentation favors SCP to produce more flavor.

Alcohol was the most abundant volatile compound in the SCP juice from the different fermentation methods, and most of the alcohols found in the fermented SCP juice were higher alcohols. Alcohols (higher alcohols) are predominantly formed by yeast from α-keto acids, either via the degradation of amino acids (valine, leucine, isoleucine, threonine, phenylalanine) by the Ehrlich pathway, or de novo biosynthesis from the carbon source [9]. Phenylethanol, which is an alcohol with a light rose aroma, isomyl alcohol, which has a floral and fruit aroma, and other flavor substances such as isobutanol, n-butanol and 2-ethxyethanol, etc., all play an important role in the aroma spectrum of SCP juices.

The myrcene and dimethyl disulfide identified in the fermented SCP juice were one of the main reasons for the unique flavor of the SCP. Compared with esters and alcohols with their high relative contents, their flavor threshold was low, so the main flavor of the SCP was greatly affected.

The results of this study not only highlighted the role of aroma-producing yeast in improving the flavor of SCP, but also laid the foundation for further study and analysis of the flavor characteristics of fermented SCP.

### 3.7. Sensory Evaluation

The sensory evaluation of SCP intuitively illustrates the flavor quality of SCP. It can be seen from Figure 6 that the aroma of the SCP experimental group fermented by mixed bacteria was higher than that of the natural fermentation group, and the aroma score of the experimental group with a mixed bacteria ratio of 1:1 was the highest. Under the conditions of low lactic acid bacteria content, less acid production, yeast can reproduce better under these conditions, and produce alcohol, ester and other flavor substances that increase the aroma of SCPs. When the content of *Lactobacilli strains* was too high, the yeast could not survive, and even died in the environment with low pH, resulting in the SCP being unable to produce more flavor aroma substances; thus, the aroma score of the mixed bacteria ratio 3:1 experimental group was lower. In addition, the taste and texture scores of the SCP in the mixed fermentation group were significantly improved compared with the natural fermentation group, but there was no significant difference in the color. The SCP fermented with the mixed bacteria ratio of 1:1 showed higher quality in aroma, taste and texture, thus improving the overall acceptability.

## 4. Conclusions

In this study, *W. anomalus* Y3 isolated from SCP fermentation broth was found to produce strong ester and wine aromas. The total ester content of the Y3 was 1.22 g/L; it could tolerate high salt concentrations (14%) and grow stably in a pH range of 3~4. The strain did not produce white film on the surface of the SCP to affect its quality, nor did strain produce cellulase and pectinase to soften the SCP. These characteristics made this strain suitable for the fermentation of SCP, and effectively improved its quality. SCP fermented with Lactobacillus plantarum at a ratio of 1:1 scored higher in the sensory evaluation. The GC–MS analysis of volatile flavor substances showed that compared with other fermentation groups, SCP with a mixed fermentation ratio of 1:1 had ethyl acetate (strong ether and fruity), phenylethanol (honey and rose), and more isobutanol, isothiocyanate, myrcene and dimethyl disulfide. According to the aroma-producing characteristics of *W. anomalus* Y3, it has broad application prospects in SCP fermentation.

## Figures and Tables

**Figure 1 foods-12-02934-f001:**
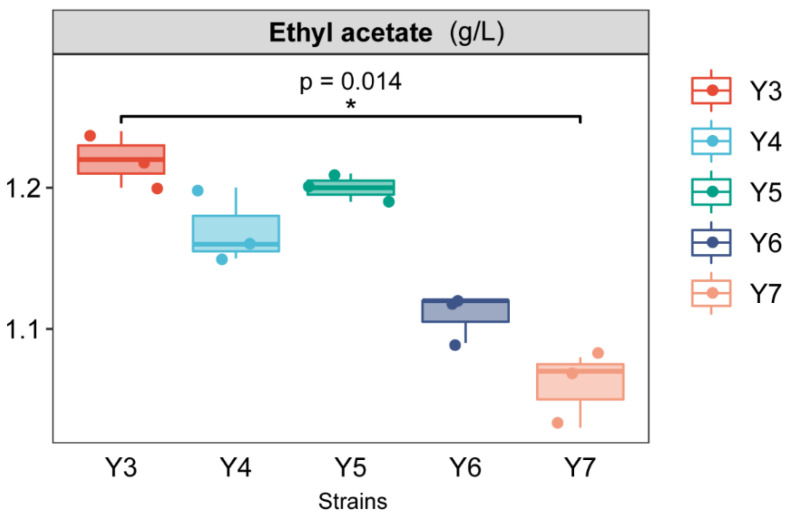
Screening results of aroma-producing yeast strains (* *p* < 0.05).

**Figure 2 foods-12-02934-f002:**
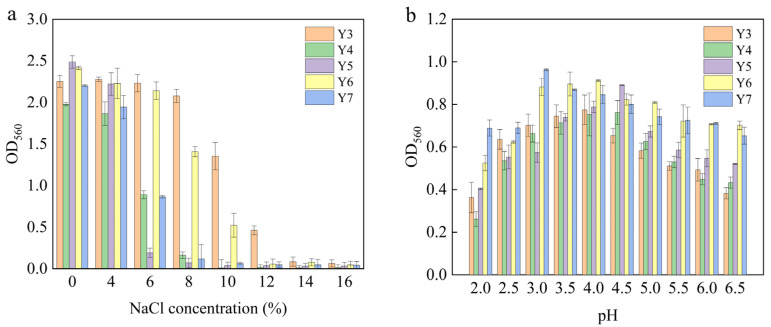
Results of growth tests at different NaCl concentrations (**a**) and pH conditions (**b**).

**Figure 3 foods-12-02934-f003:**
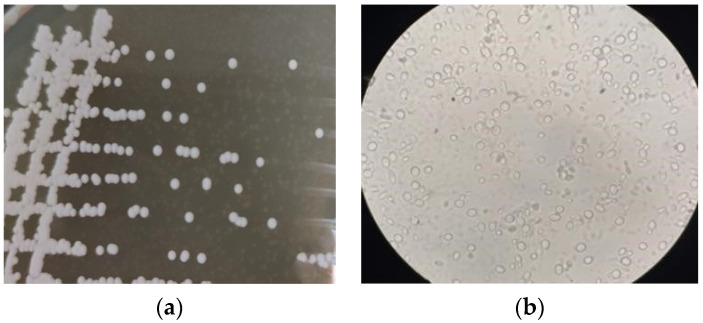
Colony morphology (**a**) and cell morphology (**b**) of Yeast Y3.

**Figure 4 foods-12-02934-f004:**
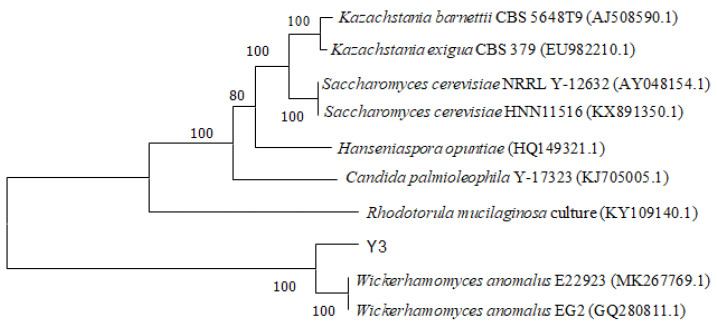
Phylogenetic tree.

**Figure 5 foods-12-02934-f005:**
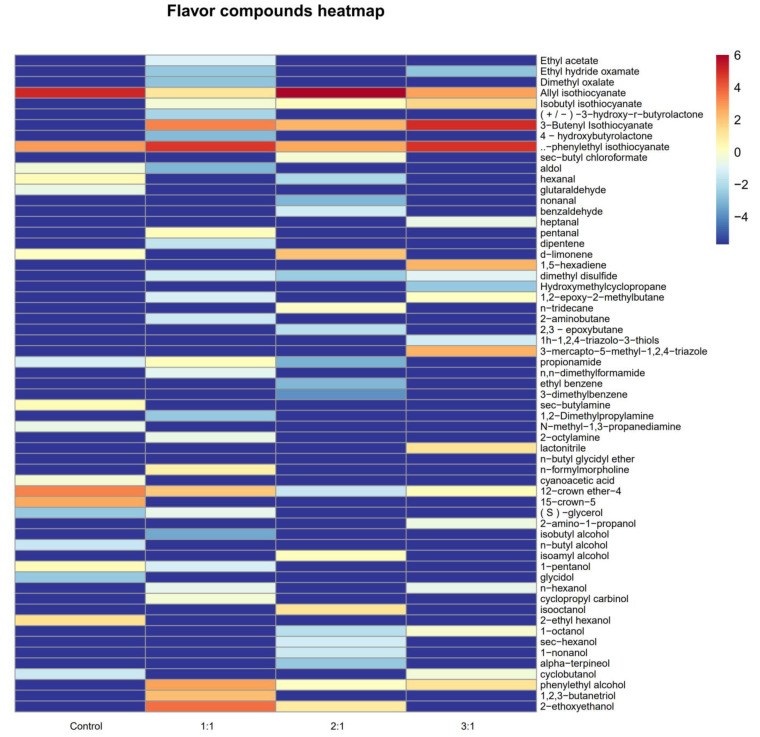
Thermal analysis of fermented SCP juice.

**Figure 6 foods-12-02934-f006:**
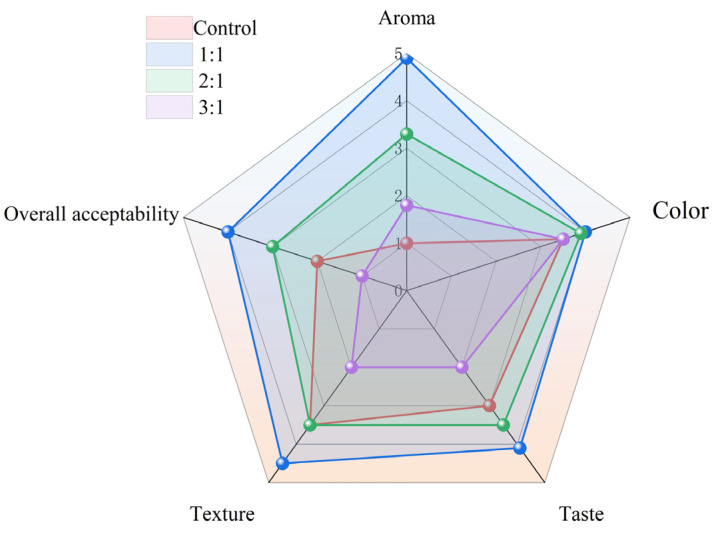
Sensory score radar chart of different proportions of *Lactiplantibacillus plantarum* and yeast-fermented SCP.

**Table 1 foods-12-02934-t001:** Preliminary screening results of aromatic yeast.

Number	Y3	Y4	Y5	Y6	Y7
Flavor characteristic	Ester, fruit and wine aromas	Ester, floral, wine aroma	Ester, fruit and wine aromas	Ester aroma, wine aroma	Ester aroma, wine aroma
Fragrance intensity	+++++	++++	+++++	+++	++++

“+” indicates fermentative flavor intensity, and more “+” symbols indicate more intense fermentative flavor.

**Table 2 foods-12-02934-t002:** Film-producing conditions.

Strains	Y3	Y4	Y5	Y6	Y7
White-colony formation	−	−	−	+	−

“−” indicates no slugs; “+” indicates slugs.

**Table 3 foods-12-02934-t003:** Cellulase and pectinase measurement screenings.

Strains	Y3	Y4	Y5	Y6	Y7
Cellulose	−	−	−	−	−
Pectinase	−	−	−	−	−

“−” indicates no enzyme production.

**Table 4 foods-12-02934-t004:** Volatile aroma components and relative contents in fermented vegetable juice.

Number	Aroma Compounds	Compound RT (min)	Relative Content (%)			
			Control	1:1	2:1	3:1
Esters						
ES1	Ethyl acetate	6.8	ND	0.48 ± 0.02	ND	ND
ES2	Ethyl hydride oxamate	10.655	ND	0.17 ± 0.03 ^a^	ND	0.151 ± 0.007 ^a^
ES3	Dimethyl oxalate	12.032	ND	0.15 ± 0.05	ND	ND
ES4	Allyl isothiocyanate	16.498	33.6 ± 0.4 ^b^	2.16 ± 0.14 ^d^	63.295 ± 3.1 ^a^	6.531 ± 0.48 ^c^
ES5	Isobutyl isothiocyanate	17.029	ND	0.89 ± 0.06 ^b^	1.178 ± 0.15 ^b^	2.987 ± 0.32 ^a^
ES6	(+/−)-3-Hydroxy-r-butyrolactone	19.144	ND	0.22 ± 0.01	ND	ND
ES7	3-Butenyl isothiocyanate	22.932	ND	9.79 ± 0.07 ^b^	5.096 ± 0.41 ^c^	32.932 ± 0.85 ^a^
ES8	4-Hydroxybutyrolactone	27.566	ND	0.127 ± 0.01	ND	ND
ES9	β-Phenylethyl isothiocyanate	37.33	7.57 ± 0.72 ^b^	27.94 ± 2.93 ^a^	6.226 ± 0.304 ^b^	29.609 ± 1.951 ^a^
ES10	Sec-butyl chloroformate	37.33	ND	ND	0.844 ± 0.09	ND
	∑		41.17	41.927	76.639	72.21
	Aldehydes					
AD1	Aldol	10.655	0.82 ± 0.055 ^a^	0.12 ± 0.017 ^b^	ND	ND
AD2	Hexanal	11.392	1.34 ± 0.27 ^a^	ND	0.23 ± 0.04 ^b^	ND
AD3	Glutaraldehyde	18.165	0.7 ± 0.006	ND	ND	ND
AD4	Nonanal	24.02	ND	ND	0.12 ± 0.011	ND
AD5	Benzaldehyde	24.02	ND	ND	0.421 ± 0.024	ND
AD6	Heptanal	26.237	ND	ND	ND	0.667 ± 0.03
AD7	Pentanal	27.566	ND	1.17 ± 0.12	ND	ND
	∑		2.86	1.29	0.771	0.667
	Alcohols					
AL1	(S)-Glycerol	8.389	0.17 ± 0.034 ^b^	0.65 ± 0.11 ^a^	ND	ND
AL2	2-Amino-1-propanol	12.032	ND	ND	ND	0.738 ± 0.05
AL3	Isobutyl alcohol	12.032	ND	0.093 ± 0.01	ND	ND
AL4	n-Butyl alcohol	13.579	0.34 ± 0.015	ND	ND	ND
AL5	Isoamyl alcohol	16.498	ND	ND	1.159 ± 0.03	ND
AL6	1-Pentanol	17.029	1.33 ± 0.06 ^a^	0.45 ± 0.06 ^b^	ND	ND
AL7	Glycidol	19.144	0.17 ± 0.01	ND	ND	ND
AL8	n-Hexanol	20.02	ND	0.62 ± 0.025 ^a^	ND	0.605 ± 0.029 ^a^
AL9	Cyclopropyl carbinol	24.02	ND	0.84 ± 0.07	ND	ND
AL10	Isooctanol	24.02	ND	ND	2.276 ± 0.23	ND
AL11	Cyclobutanol	24.02	ND	0.71 ± 0.03	ND	ND
AL12	2-Ethyl hexanol	24.02	2.42 ± 0.31	ND	ND	ND
AL13	1-Octanol	25.458	ND	ND	0.269 ± 0.002 ^b^	0.963 ± 0.012 ^a^
AL14	Sec-hexanol	26.237	ND	ND	0.423 ± 0.02	ND
AL15	1-Nonanol	27.566	ND	ND	0.396 ± 0.013	ND
AL16	Alpha-terpineol	28.829	ND	ND	0.162 ± 0.014	ND
AL17	Cyclobutanol	29.306	0.37 ± 0.02 ^b^	ND	ND	0.778 ± 0.05 ^a^
AL18	Phenylethyl alcohol	33.101	ND	6.68 ± 0.35 ^a^	1.1 ± 0.062 ^c^	2.329 ± 0.17 ^b^
AL19	1,2,3-Butanetriol	33.101	ND	4.64 ± 0.68	ND	ND
AL20	2-Ethoxyethanol	36.025	ND	12.48 ± 0.51 ^a^	2.008 ± 0.11 ^b^	ND
	∑		4.8	26.453	7.793	5.413
	Alkenes					
AK1	Myrcene	14.365	ND	0.393 ± 0.01	ND	ND
AK2	Dipentene	15.573	ND	0.33 ± 0.021	ND	ND
AK3	d-Limonene	15.573	1.1 ± 0.075 ^b^	ND	3.999 ± 0.402 ^a^	ND
AK4	1,5-Hexadiene	25.458	ND	ND	ND	5.108 ± 0.25
	∑		1.1	0.723	3.999	5.108
	Disulfide					
DI1	Dimethyl disulfide	11.392	ND	0.4 ± 0.038 ^b^	0.182 ± 0.011 ^c^	0.515 ± 0.028 ^a^
	∑		0	0.4	0.182	0.515
	Hydrocarbon					
HY1	Hydroxymethyl cyclopropane	13.579	ND	ND	ND	0.171 ± 0.017
HY2	1,2-Epoxy-2-methylbutane	17.029	ND	0.44 ± 0.039 ^b^	ND	1.09 ± 0.194 ^a^
HY3	n-Tridecane	18.165	ND	ND	0.991 ± 0.024	ND
HY4	2-Aminobutane	29.306	ND	0.37 ± 0.015	ND	ND
HY5	2,3-Epoxybutane	33.101	ND	ND	0.286 ± 0.01	ND
	∑		0	0.81	1.277	1.261
	Azoles					
AZ1	1h-1,2,4-Triazolo-3-thiols	15.573	ND	ND	ND	0.404 ± 0.02
AZ2	3-Mercapto-5-methyl-1,2,4-triazole	17.029	ND	ND	ND	5.236 ± 0.21
	∑		0	0	0	5.64
	Amides					
AM1	Propionamide	27.566	0.46 ± 0.02 ^b^	1.17 ± 0.16 ^a^	0.109 ± 0.03 ^c^	ND
AM2	n,n-Dimethylformamide	19.144	ND	0.6 ± 0.02	ND	ND
	∑		0.46	1.77	0.109	0
	Aromatics					
AR1	Ethyl benzene	13.579	ND	ND	0.126 ± 0.01	ND
AR2	3-Dimethylbenzene	13.579	ND	ND	0.067 ± 0.01	ND
	∑		0	0	0.193	0
	Amino-compound					
AC1	Sec-butylamine	10655	1.34 ± 0.13	ND	ND	ND
AC2	1,2-Dimethylpropylamine	11.392	ND	0.17 ± 0.02	ND	ND
AC3	N-methyl-1,3-propanediamine	24.02	0.69 ± 0.011	ND	ND	ND
AC4	2-Octylamine	26.237	ND	0.66 ± 0.015	ND	ND
	∑		2.03	0.83	0	0
OT1	Lactonitrile	8.389	ND	ND	ND	2.242 ± 0.31
OT2	n-Butyl glycidyl ether	12.032	ND	ND	0.019 ± 0.011	ND
OT3	n-Formylmorpholine	17.029	ND	1.69 ± 0.022	ND	ND
OT4	Cyanoacetic acid	25.458	0.88 ± 0.02	ND	ND	ND
OT5	12-Crown ether-4	33.101	10.51 ± 0.46 ^a^	3.68 ± 0.64 ^b^	0.346 ± 0.006 ^d^	1.231 ± 0.018 ^c^
OT6	15-Crown-5	33.101	6.19 ± 0.58	ND	ND	ND
	∑		17.58	5.37	0.365	3.473

“ND”: not detected; “RT”: retention time; ^a–d^ in the same line indicates significant difference at *p* < 0.05 (n = 3).

## Data Availability

The data presented in this study are available on request from the corresponding author.

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
