# Peer review of "Isolation of Aroma-Producing Wickerhamomyces anomalus Yeast and Analysis of Its Typical Flavoring Metabolites"

_foods, 2023, doi:10.3390/foods12152934_

Round 1

Reviewer 1 Report

Title

Different fonts 

Abstract

Lactobacillus plantarum -> Lactiplantibacillus plantarum

Introduction

ln. 38 membrane-producing yeast ?

ln. 38-39 The names of microorganisms has to be written in Italic till the end of the paper

Materials and methods

2.2. Isoltaion and screening of yeast strains

Please, re-write this section to become more clearly each cultivation temperature and duration.

ln. 70 Please, write something about the method for total ester determination

2.3. Performance determination of aroma-producing yeast strains

ln.74 106 ->106

ln.78  bacterial?

ln.85 CMC? There is no information for this abbreviation.

The method for determintion of reducing sugars by DNS is written very unclear, so, please rewrite it. What kind of enzyme solution? Why and for which sample you used enzyme solution?

2.4.2. Molecular biological identification

ln. 102 The target bacteria?

Results and discussion

Figure 2 -> According to me it will be better if the results are presented in a bar chart

It will be good to summarize why you have selected Y3 for identification

According to me it will be better to make sensory evaluation of all SCP produced

The quality of English is very low. 

Author Response

#Reviewer 1

  • Title: Different fonts.

Response: Title font has been modified and unified.

  • Abstract: Lactobacillus plantarum-> Lactiplantibacillus plantarum.

Response: In the abstract, Lactobacillus plantarum has been modified to Lactiplantibacillus plantarum.

  • Introduction: ln. 38 membrane-producing yeast? ln. 38-39 The names of microorganisms have to be written in Italic till the end of the paper.

Response: Considering the logical relationship before and after, the membrane producing yeast in the sentence has been deleted. And the name of microorganisms in this part and the full text have been all modified to italic writing.

  • Materials and methods: 2.2. Isolation and screening of yeast strains. Please, re-write this section to become more clearly each cultivation temperature and duration.

Response: This section has been rewritten, and each section's culture temperature and time are described in detail.

  • Materials and methods: ln. 70 Please, write something about the method for total ester determination.

Response: A detailed description of the total ester determination method has been added.

  • Materials and methods: The method for determintion of reducing sugars by DNS is written very unclear, so, please rewrite it. What kind of enzyme solution? Why and for which sample you used enzyme solution?

Response: In this study, DNS was used to determine the cellulase and pectinase produced by aroma-producing yeast, not to determine reducing sugar. Therefore, the supernatant of the yeast fermentation broth after centrifugation was used as the crude enzyme solution, and the amount of 1μg glucose or 1μg galacturonic acid produced by 1mL enzyme solution to decompose 105 substrates in 1 minute was used to define the cellulase and pectinase activity units in the aroma-producing yeast.

  • Results and discussion: Figure 2 -> According to me it will be better if the results are presented in a bar chart.

Response: Fig.2 has been changed to bar chart representation.

  • Results and discussion: It will be good to summarize why you have selected Y3 for identification.

Response Before the morphological and molecular biological identification of yeast, the explanation and summary of why Y3 yeast was selected for identification have been added.

  • Results and discussion: According to me it will be better to make sensory evaluation of all SCP produced.

Response Based on your suggestions, sensory evaluations of SCP on aroma, color, texture, taste, and overall acceptability under different fermentation conditions ( natural fermentation and mixed fermentation ) have been added to the results and discussions, and the relevant content was added in the Abstract, Introduction, Materials and methods, and Conclusions.

  • The quality of English is very low. 

Response We consulted with a native English speaker and followed his advice to revise this paper for language issues.

Other minor comments

  • 3. Performance determination of aroma-producing yeast strains. ln.74 106 ->106?

Response In the article, 106 has been modified to 106.

  • 78 bacterial?

Response In the article, bacterial suspension has been modified to yeast suspension.

  • 85 CMC? There is no information for this abbreviation.

ResponseIn determining cellulase and pectinase activity produced by aroma-producing yeast by the DNS method, CMC-Na represents sodium carboxymethyl cellulose, as explained in this paper.

  • 4.2. Molecular biological identification. ln. 102 The target bacteria?

ResponseThe target bacteria in this paper represents the screened yeast, which has been modified to the yeast strain.

Reviewer 2 Report

I attach the manuscript with my suggestions

Author Response

#Reviewer 2

  • Title: Different fonts.

Response: The title font has been modified and unified.

  • Keywords can’t be included in the title.

Response: The “aroma-producing” of keywords was deleted.

  • Materials and methods: 2.5. Liquid aroma production experiment of aroma-producing yeast. How were replicates carried out? Were compounds identified with internal standards?

Response: The volatile flavor components in SCP juice were determined by SPME-GC-MS, and the compounds were identified by the internal standard method. The 4-methyl-2-pentanol methanol solution was supplemented as the inner standard compound.

  • Results and discussion: Table 4 relative content should include SD.

Response: The relative content in Table 4 has been added SD.